

# VISIR: Technological infrastructure of an operational service for safe and efficient navigation in the Mediterranean Sea

G. Mannarini[1], G. Turrisi[1], A. D'Anca[2], M. Scalas[3], N. Pinardi[1,4,5], G. Coppini[1], F. Palermo[1], I. Carluccio[1], M. Scuro[1], S. Cretì[1], R. Lecci[1], P. Nassisi[2], and L. Tedesco[3]

[1]Fondazione CMCC – Ocean Predictions and Applications, via Augusto Imperatore 16, 73100 Lecce, Italy
[2]Fondazione CMCC – Advanced Scientific Computing, Strada Provinciale per Arnesano (complesso Ekotekne), 73100 Lecce, Italy
[3]Links Management and Technology S.p.A. – via R. Scotellaro 55, 73100 Lecce, Italy
[4]Istituto Nazionale di Geofisica e Vulcanologia, Via Donato Creti 12, 40128 Bologna, Italy
[5]Università degli Studi di Bologna, viale Berti-Pichat, 40126 Bologna, Italy

*Correspondence to:* G. Mannarini (gianandrea.mannarini@cmcc.it)

**Abstract.** VISIR (discoVerIng Safe and effIcient Routes) is an operational Decision Support System (DSS) for ship routing designed and implemented in the frame of the TESSA (TEchnology for Situational Sea Awareness) project. The system is aimed to increase safety and efficiency of navigation through the use of forecast environmental fields and route optimization. VISIR can be accessed

through both a web interface (www.visir-nav.com) and mobile applications for both iOS and Android devices. This paper focuses on the technological infrastructure developed for operating VISIR as a DSS. Its main components are described, the major challenges faced by the operational system are highlighted, and its potential for interoperability is outlined.

## 1 Introduction

Situational sea awareness through the operational distribution of oceanographic and meteorological information is a key enabler of technological applications for maritime safety and efficient transportation. In fact, the use of marine weather forecasts for route recommendations has been since long recognized (Bowditch, 2002).

However, due to the limited spatial resolution of the oceanographic forecast products, so far ap-

plications have mainly dealt with large ocean-going motor vessels or racing and leisure sailboats, mainly in a regime of open-sea navigation. In recent years, the operational availability of coastal observatories and high resolution ocean forecast products opened the way to applications even in enclosed seas and coastal waters (Proctor and Howarth, 2008). A list of both institutional and commercial ship routing systems is provided by Lu et al. (2015), and some major commercial softwares



are compared in Walther et al. (2014). A few other systems are here reviewed with an emphasis on their operational functioning.

In Montes (2005) it is reported about a model for automatizing the Optimum Track Ship Routing (OTSR) service provided by the U.S. Navy to own ships. It is a least-time routing system based on wave forecasts, whereby the safety of navigation is accounted for through speed penalty functions

related to the sea state. Interestingly, model outputs are validated versus historical records of route diversions by the U.S. war ships in the western Pacific Ocean. However, the operational system is not publicly accessible.

The Finnish transport agency developed ENSI[1], a system for providing ships navigating in the Gulf of Finland with a shore-based support. The route is first planned onboard using an ECDIS, then

it is broadcast to a Vessel Traffic Centre, where is validated against topological and marine weather information, and finally is broadcast back to the ship. Warning items are issued if the shore-based centre detects that the onboard planned route is either to cross a traffic separation scheme or to sail over shallow water.

A new concept of Sea Traffic Management (STM) has been developed during the MONALISA-

series European projects[2] and is going to be implemented in the coming years, starting with the STM Validation project[3]. In the current definition of STM, the marine voyage is the central object of analysis and development (Siwe et al., 2015). A common format and architecture for a seamless exchange of route information and voyage plans was designed and standardized during the MONALISA 2.0 project[4]. In the STM framework, voyage management services will provide support to individual

ships in both the planning process and during the voyage, also making use of route optimization services.

To the best of our knowledge, an open access, operational ship routing system, using state-of-the-art sea state and marine weather forecast for the route optimization, was still missing before we, in the frame of an Italian national project aimed at technological transfer (TESSA), developed

VISIR ([vi'zi:r], discoVerIng Safe and effIcient Routes)[5]. VISIR is meant to provide an open (not necessarily free) access, user-oriented, cross-channel system for on-demand computation of optimal routes for various kind of both motor- and sailboats navigating in the Mediterranean Sea. In VISIR-I, the first version of the system, optimization regards the total time of navigation, keeping into account safety of navigation.

In this paper we mainly report about the technological infrastructure developed for operating VISIR as a Decision Support System (DSS). However, we deem useful to first provide a compact introduction to the VISIR ship routing model in Sect.2. A detailed presentation of the operational system follows in Sect.3, and examples of functioning in Sect.4. The conclusions given in Sect.5

---

[1] https://www.ensi.fi/portal/

[2] http://monalisaproject.eu/

[3] http://stmmasterplan.com/

[4] IEC 61174, edition 4: https://webstore.iec.ch/publication/23128

[5] http://www.visir-nav.com/



summarize the authors' experience gained while realizing VISIR, and discuss the more general per-
spectives for this kind of technological infrastructure.

## 2 The ship routing model

The model behind the VISIR operational system has been developed from scratch in the frame of
the TESSA project. Its numerical structure is described in highest detail in Mannarini et al. (2015).
The model is presently coded in Matlab.

VISIR-I, the first version of the model, employs meteo-oceanographic forecast products to the end
of optimizing nautical routes. The optimization objective is the total sailing time, to be kept at an
absolute minimum, while treating safety of navigation as a constraint. Safety includes consideration
of minimum Under Keel Clearance (UKC, for both motor- and sailboats) and dynamical stability
checks (just for motorboats). VISIR-I consists of three main components, as reported in the following
subsections: the environmental forecasts, the optimization algorithm, and the vessel model.

### 2.1 Environmental forecasts

In VISIR-I both sea-state (waves) and wind forecasts are employed: the wave information is used
for motorboat and the wind information for sailboat routing.

Small and middle sized motorboats are considered by VISIR-I. In (Mannarini et al., 2015) it is
discussed what the most relevant environmental couplings for this class of vessels are, concluding
that waves are likely to be the most impacting phenomena. Thus, VISIR-I employs wave forecast
fields from an operational implementation of Wave Watch III (WW3) model in the Mediterranean
Sea (Tolman, 2009). The employed fields are significant wave height, peak wave period and wave di-
rection. They are provided with hourly resolution on a 1/16 degree (i.e., 3.75 miles in the meridional
direction) mesh. The forecasts originating from the 12:00 UTC analysis are employed.

For sailboats, 10 meter height wind forecasts from IFS model operated by the European Centre
for Medium-Range Weather Forecasts (ECMWF[6]) are employed. The model outputs are available
with 3-hourly resolution for the first 3 days after the analysis, the horizontal resolution is 1/8 degree
(7.5 miles in the meridional direction), and the forecasts refer to the 12:00 UTC analysis.

### 2.2 Optimization algorithm

VISIR-I's optimization is based on a graph-search algorithm. Dijkstra's algorithm (Dijkstra, 1959)
is chosen, for its ease of implementation and the fact that, not depending on any heuristics, it is guar-
anteed to find an optimal solution. Furthermore, since the environmental forecasts are represented
by time-dependent fields, the algorithm is modified along the guidelines by Orda and Rom (1990)
for ingesting such a dynamic information. Furthermore, VISIR's algorithm allows for voluntary ship

---

[6]http://www.ecmwf.int/en/forecasts/documentation-and-support/evolution-ifs/scorecards/scorecard-ifs-cycle-40r1



speed reduction. This option enables the ship not to change course for ensuring the vessel stability constraints, resulting in additional savings of navigational time. All the mathematical details, the algorithm's validation, and its pseudocode are provided in Mannarini et al. (2015). The target grid for the optimization is a 1/60 degree (1 mile in the meridional direction) regular mesh and the con-

nectivity of the graph is such that angles of about 27 degree can be resolved. Furthermore, VISIR is capable of avoiding the landmass even in topologic complex areas, such as peninsulas, islands, and archipelagic seas.

### 2.3 Vessel model

Two quite different approaches are adopted for modelling the dynamical response of either motor-
or sailboats to the environmental conditions, as shortly summarized in the following.

For motorboats, just displacement vessels (fishing vessels, service boats, displacement hull yachts, and small ferry boats) are considered in VISIR-I. Sustained vessel speed is obtained from a balance between the thrust provided by the propeller and various kind of resistances applied to the moving hull in any given sea state. In order to reduce the number of parameters to be set by the end-user,

the motorboat vessel model is kept simple, neglecting several mechanical effects affecting vessel dynamics, see (Mannarini et al., 2015) for more details. The six parameters to be provided by the end-user are listed in Tab.1. Furthermore, vessel stability in a seaway is considered (IMO, 2007). In particular, the dynamical conditions for the activation of three stability loss mechanisms are checked for: parametric roll, pure loss of stability, and surfriding/broaching-to (Belenky et al., 2011). Graph

edges leading, for a specific time step, to stability loss are removed from the graph previously to the run of the optimization algorithm. This way, it is ensured that optimal route does not result in an exposure to dynamical hazards.

Sailboat are described in terms of their "polar plots". These are response function expressing sailboat speed in terms of wind speed and direction. They stem from either measured sailboat perfor-

mance or so called "Velocity Prediction Programs" (de Jong et al., 2008). In VISIR-I, polar plots for a given and fixed sail are considered. Each polar plot contains a no-go-zone, accounting for the fact that direct navigation into the wind is not possible. More on this subject can be found in (Mannarini et al., accepted, 2015).

### 3  The operational system

In order to make the ship routing system of Sect.2 operational, a hardware and software infrastructure has been built in the frame of the TESSA project. In addition to VISIR, TESSA also supported the development of several other DSSs using the outputs of meteo-oceanographic models: for instance an oil spill management DSS[7] and a DSS for supporting marine search and rescue operations[8] were

---

[7]http://www.witoil.com/
[8]http://www.ocean-sar.com/



also developed. All the input meteo-oceanographic model outputs can be accessed from the Sea-
Conditions[9] portal. VISIR as an operational system inherits the infrastructural approach by TESSA,
which is a web-based architecture, exposing multichannel functionality and decoupling the service
front- and backend.

Being multichannel implies that the service is made available across different web and mobile
platforms (desktop computers, tablets, smartphones). Furthermore, the services are developed having
in mind the the "3C" paradigm (Levin, 2014). That is, the user is granted a Consistent, Continuous,
and Complementary experience, through an ecosystem of platforms where the same service is made
available. This is realized through the decoupling between service front- and backend, enabling a
thin client to drive complex data analysis on a supercomputing facility.

From a structural viewpoint, the TESSA architecture is a "matrix" of tiers and vertical appli-
cations, see Fig.1. The tiers are: the clients, the "Situational Sea-Awareness (SSA) platform", and
the "Complex Data Analysis Module" (CDAM). The vertical applications correspond instead to the
various DSSs.

The client tier allows the end-user to send commands to and receive results from the SSA-
platform. The SSA-platform bridges a bidirectional communication between the clients and the
CDAM, and provides maps of environmental fields to the client tier. The CDAM manages the in-
coming computation requests, runs the model (e.g. the one for ship routing), and returns the results.
Furthermore, an application stack distributed between the client tier and the SSA-platform provides
the end-user with the tools required for interacting with the system.

Having in mind the block diagram of Fig.2, the functioning of application stack, SSA-platform,
and CDAM is further detailed in the next three subsections, while the execution logic of the system
is documented in Sect.3.4. The section ends with a subsection describing the end-user interaction
with the system, Sect.3.5.

### 3.1    The application stack

The VISIR application stack is a system component needed to receive the end-user's requests and
deliver the ship routes to his/her device. As seen from Fig.1, the application stack consists of compo-
nents based on both the client tier and the SSA-platform. It is structured as a multi-layer architecture,
which maps the system infrastructure to the physical layer on which the application is deployed and
executed. In particular, the following three layers characterize the application stack: the presentation
layer, the business logic layer, and the data layer.

The presentation layer is in charge of the visualization through the client rendering engine. This
layer corresponds to the Graphical User Interface generally employed by desktop applications. The
VISIR presentation layer is declined into a web application and two mobile applications. The web ap-
plication (www.visir-nav.com) is a single-page application, providing universal, full-featured, cross-

---

[9]http://www.sea-conditions.com/


platform access to VISIR. It is coded in Javascript and Java. The mobile applications have been
implemented natively for each platform, i.e. in Objective C for iOS, and in Java for Android. The rationale of creating native applications is to exploit at best the hardware resources (definitively GPS, and possibly other device sensors), optimizing visual performance, and enhancing platform-specific user experience. For instance, a platform-specific type of Map Service is adopted: either MapKit Framework[10] for Apple devices, or Google Maps[11] for the web application and the Android devices. Examples of the presentation layer are provided in Sect.4. The mobile applications have been made available on the App Store and GooglePlay.

The business logic layer is the core part of the application stack and implements its logic. Its task is to receive, process and meet the incoming requests from the client. Several RESTful (Fielding, 2000)[12] services have been developed in order to manage these requests, and to forward the required information to the other components of the infrastructure. The business logic layer is also responsible for exchanging information with the data layer.

The data layer is associated to the database engine and is responsible for the data persistence (within and across sessions[13]) and their querying. It receives and fulfils the database read/write requests coming from the business logic layer. In particular, by means of this layer, all the parameters related to the ship route computation and all the results produced by the VISIR model are stored into the database of the SSA-platform.

While the presentation layer is platform-specific, the business logic layer and the data layer serve both the channels of the web application and the mobile applications.

### 3.2 The SSA-platform

The SSA-platform provides the infrastructure for processing the environmental forecast data and making the services available to public users across different channels, spanning from smartphones and web clients to (potentially) third-party Geographic Information Systems (GIS). The SSA-platform architecture can be divided into several components: a Web Portal, a Map Service, and a Message Broker, cp. Fig.2-intermediate tier. Communication between each of the clients and server-side software parts occurs according to open, standard protocols such as HTTP [14] and JSON[15].

The Web Portal is a customized web container hosting the applications, their web APIs, and providing basic shared services like authentication and user management. The VISIR-portlet (providing a universal user interface for any[16] web browser) is hosted here. The DSS-portlet is a high level software component that provides the client tier with authentication and authorization policies and

---

[10]https://developer.apple.com/library/ios/documentation/MapKit/Reference/MapKit_Framework_Reference
[11]https://developers.google.com/maps/documentation/android-api/reference
[12]http://www.ics.uci.edu/~fielding/pubs/dissertation/top.htm
[13]A session is here defined as the user activity comprised between authentication and quitting from the DSS.
[14]http://www.w3.org/Protocols/
[15]http://www.json.org/
[16]Optimized for: Firefox, Chrome, Safari.





storage of most recent computation results for later retrieval. All user credentials and permissions are tracked within the Web Portal by a User Store: this allows fine-grained permissions and access settings to the Web Portal functionalities, like usage of a specific DSS or unlocking advanced features.

The Map Service is a cross-application component (used also by the other TESSA DSSs) designed
to provide both static and dynamic maps and related functions (like data querying) to external services and applications. The Map Service is made up of a web service providing the maps, a batch rendering system updating the forecast maps on a daily basis, and a computing cluster performing the actual rendering work. The Map Service allows distribution of very large maps by delivering tiles of the environmental fields, at different scales, as a set of 256 pixels wide images ("tiles"), greatly
improving efficiency in bandwidth and client resource usage. The indexing of these images has been defined in the past by different vendors, like Google and Microsoft[17], and then standardized by the Open Geospatial Consortium in the OpenGIS Web Map Tile Service (WMTS) 1.0.0 specifications[18]. Furthermore, the dynamic maps service also provides an experimental WMTS service for improved interoperability with external GIS software. A RESTful API is provided to the clients for querying
the available maps and accessing the data browsing functions (Sect.2.1). The batch rendering system periodically fetches environmental data from the Environmental Data Storage (EDS) of the CDAM into a (SSA-platform) local EDS and triggers their initial ingestion within the system, Fig.2. The process also includes basic integrity checks and partial rendering of maps. That is, in order to save resources, the batch rendering system pre-renders just a limited set of map tiles, allowing a rapid
response for most frequently used maps. For the remaining tiles, an on-the-fly rendering is triggered, queueing the task to the computing cluster.

The Message Broker is an intermediate component between the clients and the CDAM dispatching (DSS specific) job requests on behalf of the clients (either web or mobile) to the heavy-duty computing backend, see Sect.3.3. A call-back mechanism is provided as a hook to the CDAM in or-
der to notify job completion, either successful or not, including the data payload. DSS components use this system in order to notify users about the results of their requests and perform additional actions (e.g., store the results into an historical archive for later retrieval). This software component acts as a store-and-forward queue, as it receives and stores requests, forwards them to the computing engine, and awaits a response. Clients may then retrieve the results by polling for their availability,
checking the payload, and extracting the information they require for their work. Additionally, in order to avoid an excessive load, a data retention policy can be enforced (e.g., by setting a limited time before expiration of pending requests or removing old ones).

---

[17]http://www.maptiler.org/google-maps-coordinates-tile-bounds-projection/
[18]http://www.opengeospatial.org/standards/wmts



### 3.3 The Complex Data Analysis Module (CDAM)

The Complex Data Analysis Module or CDAM enables advanced data processing on the datasets
produced by the meteorological and oceanographic models used within the TESSA project. Specifi-
cally, it represents a back-end connector between the client oriented services, deployed on the SSA-
platform, and the model execution related to a specific DSS such as VISIR (D'Anca et al., 2016, to
be submitted).

In this perspective, the CDAM was designed and developed in order to hide the internal complex-
ity of the underlying ship routing model and ease the submission of the execution and the retrieval of
the results by the upper layers of the operational chain. Moreover, a high modularity implements the
separation of concerns, while the adoption of standard interfaces and existing technologies, such as
JSON format and REST web services, ensures flexibility and interoperability of the software compo-
nents. Finally, in order to guarantee a high security level, the incoming requests rely on the HTTPS[19]
protocol, while a dedicated private network channel has been setup for the submission of VISIR jobs
on the target computing infrastructure.

From a technological viewpoint, a two-layer logical architecture has been implemented (see Fig.2-
lower tier). The first layer, the CDAM-Gateway, has been designed to be the entry point for job
submissions; it is responsible for managing the incoming requests and for interfacing with the target
infrastructure for the algorithm execution. The second layer, the CDAM-Launcher, has the responsi-
bility to perform the submission and to correctly manage the job execution.

Delving into more detail of the CDAM-Gateway, it consists of two modules: the RESTful web
interface and the CDAM-Scheduler. The former provides the SSA-platform with an uniform REST
interface for accepting and managing the job execution requests provided in a JSON format. The
latter sets the VISIR execution environment and forwards the correct input parameters to the CDAM-
Launcher.

The CDAM-Launcher is hosted on the target computing infrastructure and is responsible for prop-
erly managing the job submission. It relies on the workload manager SLURM[20] to perform the ex-
ecution in a cluster environment. As in the case of the CDAM-Scheduler, the launcher provides a
specific module for the VISIR submission management. From an operational point of view, an in-
coming request for an execution of the VISIR model triggers the following sequence of operations on
the CDAM-Launcher side: i) check the parameters received from the CDAM-Scheduler; ii) prepare
the input files needed by the model; iii) launch the model execution; iv) once the job is completed,
contact the SSA-platform; v) send back the results.
Following the same approach taken for the job request, the results are embedded into a JSON
object compliant with the specific JSON schema defined for the CDAM.

---

[19]https://tools.ietf.org/html/rfc2818
[20]http://slurm.schedmd.com/



### 3.4 Execution logic

For the description of the logical execution of VISIR, we hereafter refer to the numbered steps in the sequence diagram of Fig.3.

The access to VISIR is open after authentication (1). At the moment, however, also a valid subscription is required in order to run the route computation service. The authentication is then granted by the DSS-portlet hosted on the SSA-platform (2). At this point, after departure and arrival location for the route have been set, a route computation request can be submitted (3). Once it is forwarded through the DSS-portlet (3') to the Message Broker, a correlation-ID is provided back to the DSS-

portlet (4a). At the same time, the Message Broker forwards the request to the CDAM (4b). While the VISIR model is run on the supercomputing facility controlled by the CDAM (5b), a polling activity starts at the client (5a) and is forwarded down to the level of the Message Broker (5a'). Such a querying is iterated till the Message Broker has received the model results from the CDAM (6). They are immediately propagated up to the level of the DSS-portlet (7) and, finally, to the client (8).

The different logical functions played by the clients, the DSS-portlet, the Message Broker, and the CDAM, enable a physical separation of the assets of the system. This is indeed the case, since the TESSA architecture mirrors the organizational structure of the TESSA consortium.

### 3.5 Use of the system

The end-user experience of VISIR entirely occurs within the presentation layer of the application

stack. Its main elements are: a menu, a geographic map, and a linechart[21]. Upon authentication, the submission of a route computation can be completed by a minimum of three clicks. The left menu offers three modality to enter the two locations between which the route has to be computed: i) by typing their lat/lon coordinates; ii) by clicking/tapping on their positions on the map; iii) by typing the name of a land-based location. The latter option exploits, depending on the platform, a Google

or MapKit Framework API for georeferencing toponyms (e.g. "Otranto" is mapped to (40.14390 N, 18.49117 E) ).

A crucial point is subsetting the domain used by the algorithm for computing the optimal route. By default, this domain corresponds to a box whereby two opposite vertexes are the route endpoints. However, due to specific domain topology (peninsular or archipelagic regions) and environmental

conditions (leading to unsafe navigation), a suboptimal route or no route at all may result from using the default bounding box. Thus, we designed all the interfaces in a way that the bounding box can be interactively resized. Furthermore, if a land-based endpoint is selected (such as "Lecce"), the next sea position within the bounding box and with positive UKC (i.e., sea depth larger than vessel draught) is automatically retrieved by the model. Three pre-defined sets of motorboat parameters (a

displacement hull cabin cruiser, a fishing vessel, and a small ferryboat) can be selected and edited.

---

[21]At present, the linechart is just available for the web application (v.4.1).



Alternatively, a sailboat type can be picked up from a database of boats with lengths between 7 and 18 meter. From the "Advanced Settings" section of the left menu (in case of motorboat) the individual safety constraints can be checked and the voluntary speed reduction can be optioned. For both motor- and sailboat, it is also possible to set a minimum offshore distance to be met by each
route waypoint.

Upon route submission, a progress bar with an estimate of the waiting time appears. Following the features of the algorithm, Sect.2.2, such a time depends on both the bounding box area and the duration of the ship route. The random availability of the internet band and the unpredictable load of the computing resources do not allow to provide the end-user with an exact estimation of such
waiting time. This is why it is provided by the VISIR dialog window together with an uncertainty, whose value is identified by means of empirical tests, like those provided in Fig.4. From a comparison between the waiting time found by a VISIR end-user and the bare algorithm running time, it is apparent that, for short routes (below 100 NM), the waiting time approaches an offset of a few seconds, while the time spent in the optimization algorithm scales down with a power law. Thus, the
offset time is attributed to model pre-processing tasks and the bi-directional communications among the system components (steps 1.-3. and 6.-8. in Fig.3). For longer routes instead, both the waiting time and the algorithm performance increase with a power law of the route length. Fitted coefficients of such trends are reported in Tab.3.

If the route computation fails, a diagnostic message appears, with a hint to what should be changed
in order to get a successful result. We list a few of such diagnostic messages in Tab.2. If the route computation succeeds, both a geodetic and an optimal route are displayed on the map. The geodetic route is a least-time route not taking into account the dynamic environmental conditions, but still complying with the static safety constraints related to the bathymetry and the shoreline. The optimal route takes into account the dynamic environmental conditions both to the end of computing vessel
kinematics and, just for motorboats, also for avoiding the dynamical hazards of navigation (see Sect.2.3).

## 4 Examples

In order to demonstrate the operational functioning of VISIR, we report in this section about two execution scenarios, one for motor- and the other one for sailboat.

### 4.1 Motorboat route

In Fig.5 a fishing vessel route in the Aegean Sea is displayed. The geodetic route connects departure and arrival location while skipping the islands in between. The optimal route, besides avoiding the shoreline, contains a significant eastbound diversion (Fig.5a). This is instrumental in avoiding the



rough seas experienced along the geodetic route. The resulting optimal route, though 24 miles longer, is nearly 1 hour faster than the geodetic one.

The VISIR web application allows displaying on the geographic map various combination of forecast fields. In Fig.5a the peak wave period and wave direction fields are shown. Furthermore, the linechart below the map can display various kinematical and environmental fields along the routes (Fig.5b,c). Each waypoint on the map is bijectively linked to the corresponding waypoint on the linechart (note highlighted waypoints in Fig.5a,b), enabling an immediate georeferencing of linechart information, or, inversely, temporal localization of a waypoint. Furthermore, upon querying a waypoint (either from the map or from the linechart), the map of the forecast field at that specific time step is displayed, highlighting the time-dependent information processed by the VISIR model. In Fig.5b the vessel Speed Over Ground (SOG) for both routes is compared to the top speed of the vessel in calm weather conditions. It is seen that the optimal route, sailing in calmer seas, enables the vessel's propulsion system to sustain a larger SOG. The route performance information are summarised in the route register in the left menu. As another option, the safety indexes for vessel stability discussed in Mannarini et al. (2015) can be displayed in the linechart.

In Fig.5 the same route on top of the significant wave height and wave direction fields is shown, as it is visualized on an iPad (d) and an iPhone (e) screen. In fact, the MapKit Framework cartography and the possibility to rotate the geographical Nord of the map can be noted.

### 4.2 Sailboat route

First of all, we should stress the fact that, due to the limited capability of producing thrust through wind, a sailboat route between given endpoints is not always feasible. In particular, there are limitations for upwind motion and for too weak or too strong wind intensities (Mannarini et al., accepted, 2015). For this reason, it is even less likely that the geodetic route - the spatially shortest path between the endpoints- is feasible. Thus, in VISIR-I the sailboat geodetic route is provided just as a topological information, without any reference to its kinematical aspects. Furthermore, the end-user is informed about the possible difficulty in computing the route by an info box including an hint to check the polar plot parameters (Fig.6a').

In Fig.6, a First 36.7 (a 11 meter long boat) route around the island of Crete is displayed. In the actual case shown, it is seen that the geodetic and the optimal route sail on opposite sides of the island (Fig.6a,d,e). This is suggested by the algorithm in order to avoid the dead calm areas in the lee of the high (> 2000 meter a.s.l.) mountains of Crete. The linecharts selected for this case study show the Speed Over Ground (SOG) and the True Wind Angle (TWA) with respect to cumulative time and date/time since departure, respectively. The SOG linechart (Fig.6b) also includes the Velocity Made good to Course (VMC) information, demonstrating that the initial eastbound diversion of the optimal route locally implies departing from the target (VMC < 0). The TWA linechart (Fig.6c) shows that the algorithm suggests the boat to sail at a TWA close to the minimum possible for the actual polar



plot during the close-hauled phase (upwind) and at a TWA close to the maximum possible during the broad reach phase (downwind).

The sailboat routing is at the moment available, besides on the web application, just on a development version of the Android mobile application, whose screenshots for tablet (Fig.6d) and smartphone (Fig.6e) are anticipated in this paper.

**5 Conclusions**

Starting from the VISIR model, a DSS for on-demand computation of optimal ship routes in the Mediterranean Sea has been put into operations. It addresses displacement hull motorboats and sailboats of various sizes. Shoreline and bathymetry are the static databases used for checking the minimal requirements for the safety of navigation. Forecasts of sea state and wind from third party

providers represent the dynamical information used for the minimization of the route duration and for checking vessel stability. The operational system is cross-channel available, and client applications for the web browser and iOS an Android devices have been developed.

The infrastructure developed for running VISIR in an operational way is to a large extent general, and it was successfully employed as a template for other DSSs developed within the TESSA project.

One of the most significant authors' experience is that the realization of the VISIR operational system (2013-2015) also had a positive feedback on the initial phase (2012-2014) of development of the VISIR ship routing model. This was due to the fact that the operational service required the model to reach a level of robustness far beyond the case study testing. Exceptional and challenging environmental conditions highlighted issues and bugs of the model, and the demanding computa-

tional requirements of a multi-user systems pushed for a more and more efficient model code.

Numerous developments of the VISIR ship routing model are envisioned and some of them were already discussed in (Mannarini et al., 2015). As next development possibilities for the operational infrastructure, we foresee the realization of a "VISIR web API" for granting interoperability with third party softwares. Also, a route caching system or a desktop application could be a viable solution

for using the system in the absence of internet connectivity. Interoperability along the lines of the new STM concept is possible, in particular the routes computed by VISIR can be made compliant with the route exchange format recently approved as a standard for ECDIS.

*Acknowledgements.* Funding through TESSA (PON01_02823/2) project is gratefully acknowledged. Yogesh Kumkar and Andrea Villani are thanked for their initial contribution to the development of the operational

infrastructure and the VISIR mobile application for Android, respectively. Roberto Bonarelli provided an appreciated feedback on the sailboat interface.





**Table 1.** Parameters of the motorboat model and their values for the case study of Fig.5.

| symbol | name | units | typical value |
|--------|------|-------|---------------|
| $P$ | actually delivered engine power | hp | 650 |
| $c$ | top speed | kn | 10.7 |
| $L$ | length at the waterline | m | 22 |
| $B$ | beam (width at waterline) | m | 6 |
| $T$ | draught | m | 2 |
| $T_R$ | natural roll period | s | 5.4 |

**Table 2.** Examples of diagnostic messages output by the VISIR-I user interface in case of failure of the route computation and related hints for fixing it.

| error code | diagnosis | hint | |
|------------|-----------|------|---|
| | | motorboats | sailboats |
| 44 | no geodetic route found | enlarge bounding box buffer zone or move extreme points further offshore or reduce value of minimum offshore distance | *[the same]* |
| 43 | no route found | enlarge bounding box buffer zone or relax safety constraints | enlarge bounding box buffer zone or change sailboat class |
| 33 | bounding box too large ! | reduce distance between departure and arrival point or reduce extent of bounding box buffer-zone | *[the same]* |
| 16 | bounding box too small or too narrow ! | increase distance between departure and arrival location or enlarge buffer zone of bounding box | *[the same]* |

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



Natural Hazards
and Earth System
**Table 3.** Parameters and scores for the least square fits shown in Fig.4. The fits of the UI waiting time are of type $cx^d + e$, while the Dijkstra's algorithm ones are of type $cx^d$.

|  |  | [units] | motorboat | sailboat |
|---|---|---|---|---|
| UI waiting time | $c$ | [sec·NM$^{-d}$] | $3.5 \cdot 10^{-6}$ | $5.8 \cdot 10^{-5}$ |
|  | $d$ | [-] | 2.95 | 2.46 |
|  | $e$ | [sec] | 17.8 | 10.5 |
|  | $R^2$ | [%] | 97 | 96 |
| Dijkstra's algorithm | $c$ | [sec·NM$^{-d}$] | $4.1 \cdot 10^{-6}$ | $5.6 \cdot 10^{-6}$ |
|  | $d$ | [-] | 2.63 | 2.58 |
|  | $R^2$ | [%] | 95 | 94 |

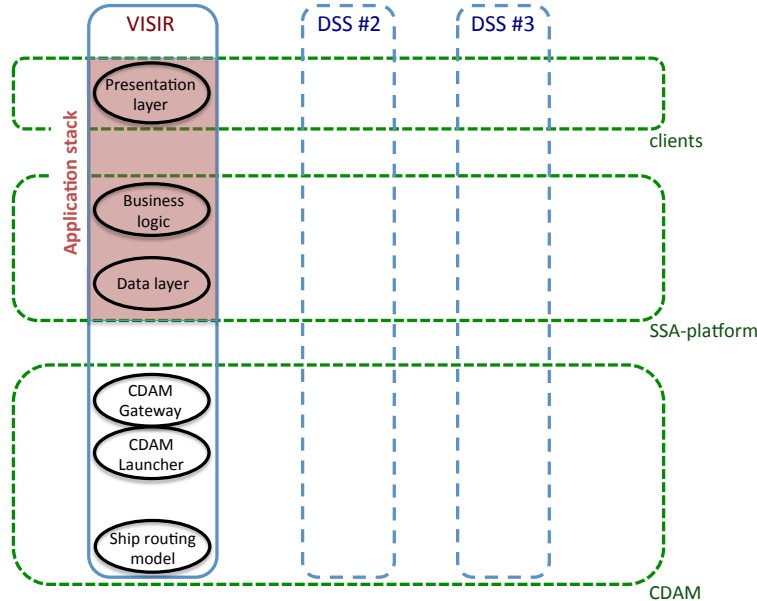

**Figure 1.** "TESSA matrix": the three horizontal tiers and the vertical applications or DSSs. The red shaded area is the VISIR application stack.

D'Anca, A., Nassisi, P., Palazzo, C., Conte, L., Lecci, R., Cretì, S., Mannarini, G., Coppini, G., Fiore, S., Pinardi, N., and Aloisio, G.: A multi-service data management platform for scientific oceanographic products, Nat. Hazards Earth Syst. Sci. Discuss., 2016, to be submitted.

de Jong, P., Katgert, M., and Keuning, L.: The development of a Velocity Prediction Program for traditional Dutch sailing vessels of the type Skûtsje, in: 20th HISWA Symposium, 2008.

Dijkstra, E. W.: A note on two problems in connexion with graphs, Numerische mathematik, 1.1, 269–271, 1959.





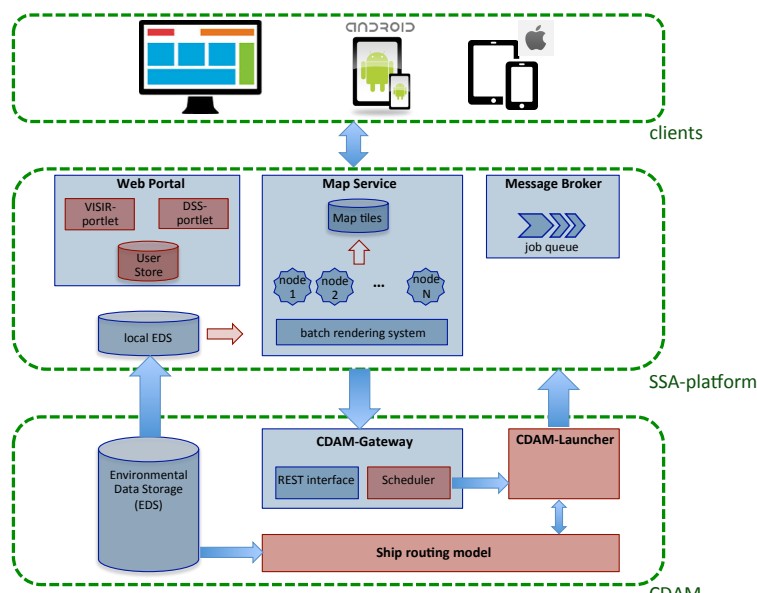

**Figure 2.** Functional structure of the TESSA system. Red boxes refer to VISIR-specific components hosted on the TESSA infrastructure (blue). Cylinders represent databases; boxes are services; icosagons are computational nodes.

Fielding, R. T.: Architectural styles and the design of network-based software architectures, Ph.D. thesis, Uni-
versity of California, Irvine, 2000.

IMO, M.: 1/Circ. 1228 Revised guidance to the Master for avoiding dangerous situations in adverse weather
and sea conditions, International Maritime Organization (IMO), London, UK, 2007.

Levin, M.: Designing Multi-device Experiences: An Ecosystem Approach to User Experiences Across Devices,
"O'Reilly Media, Inc.", 2014.

Lu, R., Turan, O., Boulougouris, E., Banks, C., and Incecik, A.: A semi-empirical ship operational performance
prediction model for voyage optimization towards energy efficient shipping, Ocean Engineering, 2015.

Mannarini, G., Pinardi, N., Coppini, G., Oddo, P., and Iafrati, A.: VISIR-I: small vessels, least-time nautical
routes using wave forecasts, Geosci. Model Dev. Discuss., 8, 7911–7981, 2015.

Mannarini, G., Lecci, R., and Coppini, G.: Introducing sailboats into ship routing system VISIR, IEEE Xplore,
accepted, 2015.

Montes, A. A.: Network shortest path application for optimum track ship routing, Ph.D. thesis, Naval Postgrad-
uate School, Monterey, California, 2005.

Orda, A. and Rom, R.: Shortest-path and Minimum-delay Algorithms in Networks with Time-dependent Edge-
length, J. ACM, 37, 607–625, 1990.

Proctor, R. and Howarth, M.: Coastal Observatories and operational oceanography: a European perspective,
Mar. Technol. Soc. J, 42, 10–13, 2008.





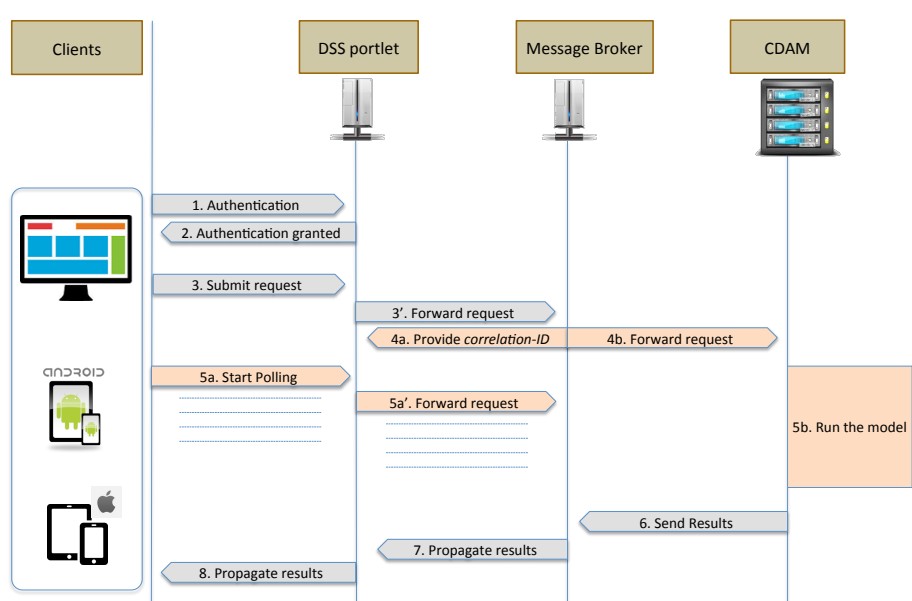

**Figure 3.** Data flux diagram. The downward oriented vertical coordinate is the time elapsed since the user has started interacting with VISIR, while the horizontal coordinate goes from high-level to low-level operations. Orange shading indicate simultaneously occuring operations. The dotted lines indicate replicas of steps (5a) and (5a'), performed till CDAM results are made available to the Message Broker.

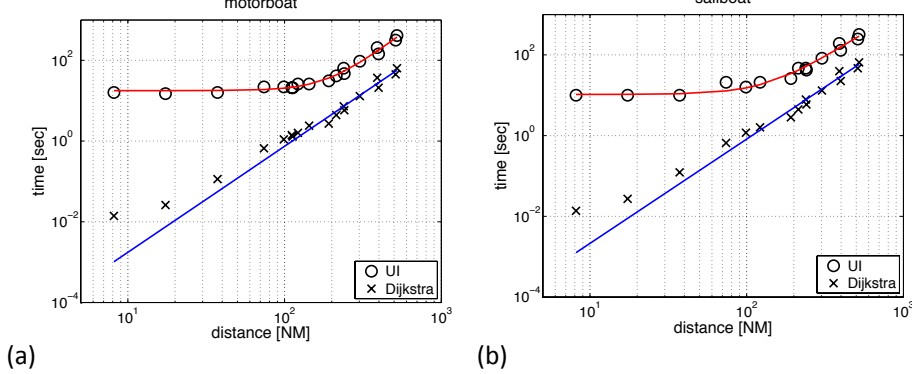

**Figure 4.** Performance of VISIR operational system. Panels (a) and (b) refer to the motorboat and sailboat submodel respectively. In each panel, the time required for the execution of a job submitted from the User Interface ("UI") is compared to the time spend by the model in the computation of the optimal route via the Dijkstra's algorithm ("Dijkstra"). Experimental results (markers) are fitted by power law functions of the type $cx^d + e$ for UI (red lines) and $cx^d$ for Dijkstra (blue lines). See Tab.3 for values of the fitted parameters.

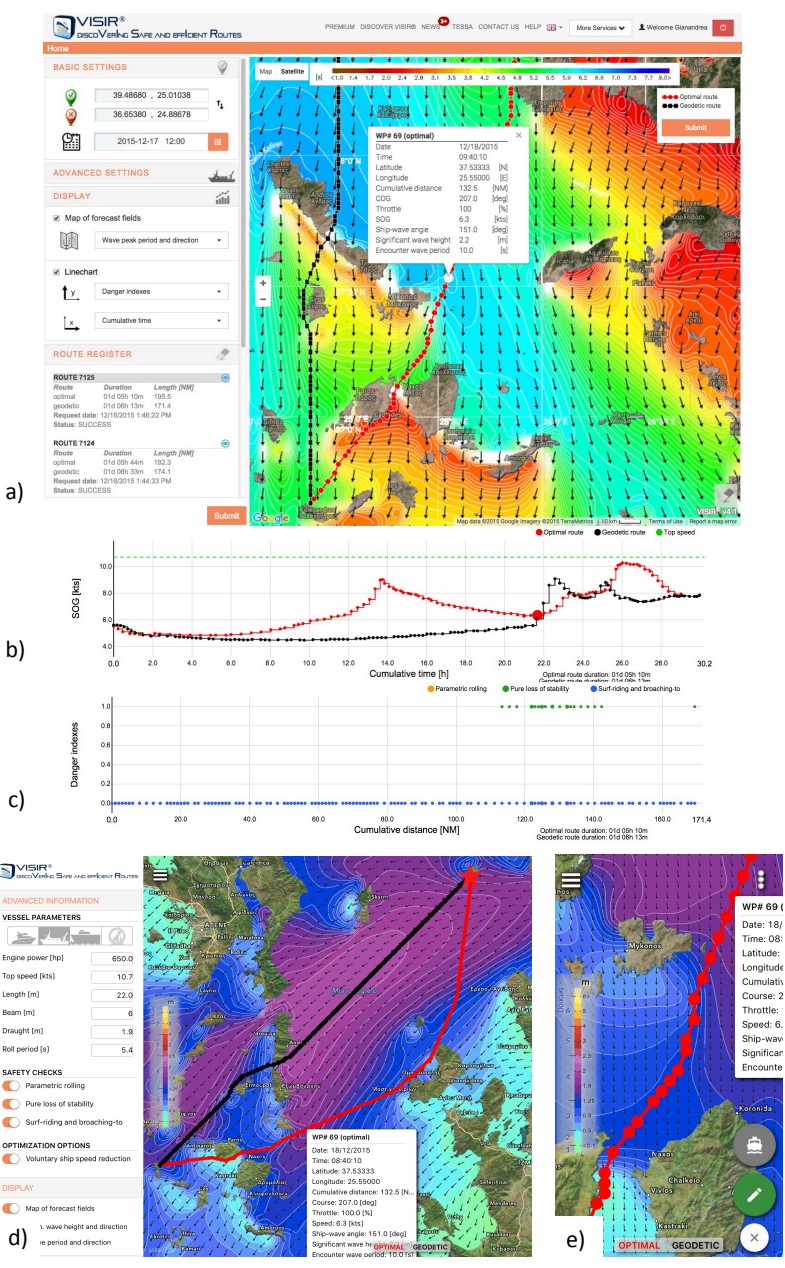

**Figure 5.** A motorboat route computed by VISIR. A displacement vessel with parameters as in Tab.1 is employed for the computations. Panels a), b), c) belong to the web application; panels d) and e) to the iOS mobile app respectively on an iPad and iPhone. The shaded field in a) is the peak wave period and wave direction; while the shaded field in d) and e) is significant wave height and wave direction. Panels b) and c) are two possible visualisations of the linechart below the map of the web application: in b) the SOG vs. cumulative time since departure and in c) the danger index vs. cumulative distance since departure is visualised.

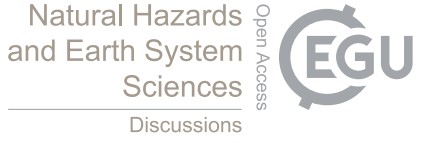

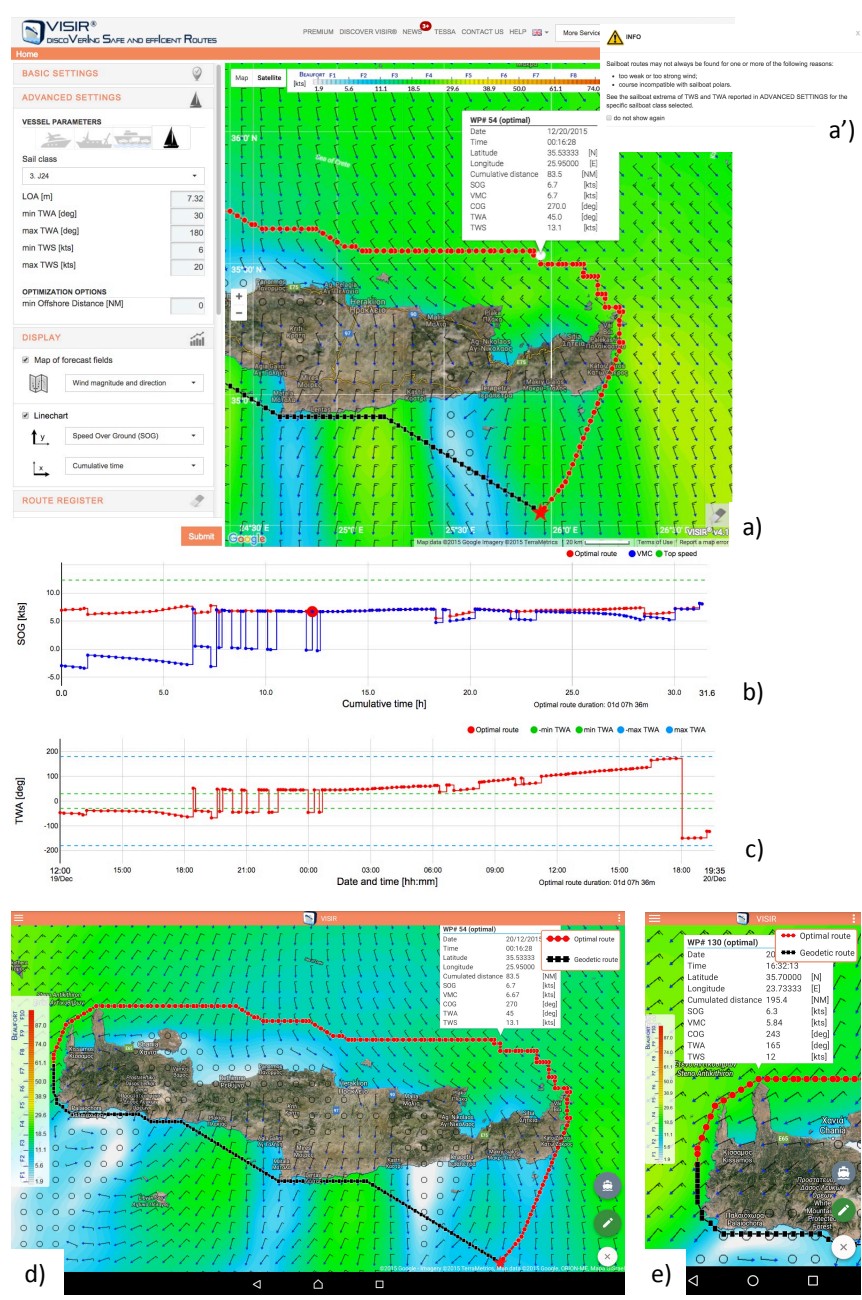

**Figure 6.** A sailboat route computed by VISIR. The shaded field in the maps is the 10 meter height wind intensity and its direction. A "First 36.7" sailboat is employed for the computations. Panels a), a'), b), c) belong to the web application; panels d) and e) to the Android mobile app respectively on a tablet and a smartphone. Panels b) and c) are two possible visualisations of the linechart below the map of the web application: in b) the SOG vs. cumulative time since departure and in c) the TWA vs. date and time is visualised.



Siwe, U., Lind, M., Hägg, M., Dalén, A., Brödje, A., Watson, R., Haraldson, S., and Holmberg, P.-E.: Sea Traffic Management - Concepts and Components, in: 14th International Conference on Computer and IT Applications in the Maritime Industries, Ulrichshusen, 11-13 May 2015, edited by Volker, B., pp. 281–289,

Technische Universität Hamburg- Harburg, 2015.

Tolman, H. L.: User manual and system documentation of WAVEWATCH III TM version 3.14, Technical note, MMAB Contribution, 2009.

Walther, L., Burmeister, H.-C., and Bruhn, W.: Safe and Efficient Autonomous Navigation with Regards to Weather, in: 13th International Conference on Computer and IT Applications in the Maritime Industries,

Redworth, 12-14 May 2014, edited by Volker, B., pp. 303–317, Technische Universität Hamburg- Harburg, 2014.