# Peer review of "VISIR: Technological infrastructure of an operational service for safe and efficient navigation in the Mediterranean Sea"

_Natural Hazards and Earth System Sciences, 2016_

## Referee Comment (RC1) · C. Granell (Referee) · 13 Feb 2016

The paper describes a decision support system (DSS) for on-demand computation of optimal routes ship routes in the Mediterranean sea. The focus of the paper is not on the modelling part but on the operational infrastructure and how the set of software components are brought together to realise the DSS. In this sense, the paper clearly describes the objective, role and scope of each software component and how this relates to each other. Such technical description is later accompanied by a couple of examples to demonstrate the operational functioning of the system.

Even though some of the client apps devised by the authors are in progress, it is quite clear that the core contribution of the paper is not these client application but the multi-

layered architecture that supports asynchronous computation of optimal routes (the model itself). Developed web services adhere to well-known web standards and best practices. the heart of the system seems to be the message-based broker because, to this reviewer , it is intended to make the system more scalable (not mentioned by authors). As there are so many pieces connected acting asynchronously, I am wondering about performance issues of the system. These aspects are not presented or discussed in the paper, so I encourage then the authors to specify limitations or performance penalties the system has, either by creating a new section to this matter or by extending sect 4.1 and 4.2 accordingly.

Please also revise the citation style of Mannariani et al. (2015) in the body of the paper because I was not able to distinguish which one *(of two in the reference list) the authors actually referred to.

In summary I found the paper quite interesting and technically sound. The topic of the paper, and the design and architecture of the system, is acceptable. As the paper focuses on the overall functioning of the system, and how the distinct pieces are connected, more details on the assessment methods and performance evaluation are required to truly demonstrate the system.

---

## Referee Comment (RC2) · Anonymous Referee #2 · 20 Apr 2016

I have just finished reviewing the manuscript titled "VISIR: Technological infrastructure of an operational service for safe and efficient navigation in the Mediterranean Sea", submitted for publication in NHESS. The authors present the software and operational architecture of a DSS for safe and efficient navigation. There is no doubt that the work is of high standard and involves large-scale operational modeling of several physical parameters, combined with a multi layer interface for post-processing and visualization. I also found that the online platform is of good quality and rather intuitive. Finally the paper is well written, but I am afraid I cannot recommend publication without major changes. My main concern is about the content of the paper, and the selection of the journal, even though it is submitted as part of a special issue on situational sea

awareness. The paper is focusing on the software platforms and technology used to calculate and visualize optimal routes. Therefore it feels more like a technical report or manual written by a software developer and not like a manuscript prepared for a natural hazards journal. No natural processes are mentioned, while there is no theory, no initial hypothesis, no analysis and no results. The manuscript mentions some technical aspects of a web application and even though I lack the expertise to follow and evaluate the system architecture, the description looks rather vague so it doesn't really provide the reader with a clear idea of how the system was built. The examples provided in the end are the most relevant part of the paper to the scope of the journal and the special issue, but no background on the route evaluation method is provided and my final impression was that the work was submitted to the wrong journal. Starting from the title, I see a weak link between 'safe and efficient navigation' and natural hazards. Acknowledging that the title is relevant to the special issue, I find that the authors should move the software aspects to the supplementary material and focus more on the natural processes involved, as well as on the best route calculation methodology. The title of the special issue is also the only reason I didn't recommend to reject the manuscript, but it is clear that as a coastal oceanographer I don't have the right background to review the work, therefore I would prefer not to review the revision. Finally, the best route functionality is not available in the free version of the online system, therefore I would recommend that reviewers should be provided with free online access to the full system.
* * *

---

## Referee Comment (RC3) · Anonymous Referee #2 · 23 May 2016

After connecting to the online platform using the user credentials provided I can provide some more feedback about the system presented. VISIR is an impressive platform with several components, while its interface is user friendly and intuitive. From that aspect I would consider it as an example of how science could provide services to the society. Some issues which could make the platform even more user-friendly are: (1) every time I was switching from one service to the other, I was asked to log in again; which in theory shouldn't be necessary; (2) VISIR was often not generating a route due to problems with the calculation area, automatically generated based on the origin and destination point. If I understood correctly that box sometimes can include land areas which are blocking sea access and the calculations fail. Then the user is

asked to extend the bounding box something that makes the application less intuitive, and I assume there must be a way that the correct boundaries automatically so that an outcome is always generated. Apart from the application which is impressive, I insist on my previous recommendation that a manuscript published in NHESS should present, evaluate and discuss the scientific approach the routes are calculated, rather than the IT systems and environments the web interface was built.

————————————————

---

## Author Comment (AC1) · 23 Jun 2016

(In the following "NHESSD" stands for the paper and under Review. When not specified, all other references to equations, figures, and tables are relative to the present document.)

**General comments**

*The paper describes a decision support system (DSS) for on-demand computation of optimal ship routes in the Mediterranean sea. The focus of the paper is not on the mod-*

*elling part but on the operational infrastructure and how the set of software components are brought together to realise the DSS. In this sense, the paper clearly describes the objective, role and scope of each software component and how this relates to each other. Such technical description is later accompanied by a couple of examples to demonstrate the operational functioning of the system. Even though some of the client apps devised by the authors are in progress, it is quite clear that the core contribution of the paper is not these client application but the multi-layered architecture that supports asynchronous computation of optimal routes (the model itself). Developed web services adhere to well-known web standards and best practices. [..] In summary I found the paper quite interesting and technically sound. The topic of the paper, and the design and architecture of the system, is acceptable.*

*–Authors' response:*
The Referee correctly focused on the manuscript's scope and we are glad to see that our effort to adhere to the state-of-the-art of technology was acknowledged.

*As the paper focuses on the overall functioning of the system, and how the distinct pieces are connected, more details on the assessment methods and performance evaluation are required to truly demonstrate the system..*

*–Authors' response:*
 See specific comment **B** below.

**Specific comments**

**A** - *The heart of the system seems to be the message-based broker because, to this reviewer, it is intended to make the system more scalable (not mentioned by authors).*

*–Authors' response:*
The Message Broker (MB) certainly has a central position within the TESSA architecture, in the sense that it brokers between the client-oriented services and the backed model computations. However, each individual VISIR system component owns a given level of scalability; not just the MB, but also the CDAM Gateway and the cluster infrastructure that are managed by the slurm scheduler.

*–Authors' changes to manuscript:*
A1.
A1) On P5, row 139, to insert:
"Each individual VISIR system component owns a given level of scalability; not just the MB, but also the CDAM Gateway and the cluster infrastructure that are managed by the slurm scheduler."

**B** - *As there are so many pieces connected acting asynchronously, I am wondering about performance issues of the system. These aspects are not presented or discussed in the paper, so I encourage the authors to specify limitations or performance penalties the system has, either by creating a new section to this matter or by extending sect 4.1 and 4.2 accordingly.*

*–Authors' response:*
We thank the Referee for this interesting observation. It will lead to a significant improvement of the manuscript.

We actually already provided some hint about the performance of the VISIR operational system in Tab.3 and Fig.4 of NHESSD. However, following the Referee's question, we made new extensive experimental tests for augmenting that information, see "Authors' changes to manuscript" below.

*–Authors' changes to manuscript:*
B1, B2, B3, B4, B5, B6, B7, B8, B9, B10, B11, B12, B13, B14.

B1) In Abstract, starting from row 6, replace with:
"This paper focuses on the technological infrastructure developed for operating VISIR as a DSS. Its main components are described, the performance of the operational system is assessed through experimental measurements, and a few case studies are presented."

B2) In Introduction, starting from P2 row 53, replace with:
"A detailed presentation of the operational system follows in Sect.3, and its functioning and performance are addressed in Sect.4. A few case studies are discussed in Sect.5, before Conclusions are drawn in Sect.6."

B3) On P9, edit Sect. 3.4 "Execution logic" starting from row 259 as follows:
"Once it is forwarded through the DSS-portlet (3') to the Message Broker (MB), a correlation-ID is provided back to the DSS-portlet (4a) and the client application (4b). At the same time, the MB forwards the request to the CDAM (4c). While the VISIR model is run on the supercomputing facility controlled by the CDAM (6 a-c), a polling activity starts at the client (5a) and is forwarded down to the level of the MB (5a'). Such a querying is iterated until the model results from the CDAM are sent to the MB (7). They are immediately propagated up to the level of the DSS-portlet (8) and, finally, to the client (9)."

B4) Rename NHESSD Section 3 into "Operational infrastructure"

B5) Define a new Section 4 "Functioning of the operational system" and renumber NHESSD Sections 4 and 5 to Sections 5 and 6 respectively.

B6) Renumber Sect. 3.4 to Sect. 4.1 "Execution logic" and Sect. 3.5 to Sect. 4.3 "Use of the system"

B7) On P9, insert a new Sect. 4.2 "System Performance" as follows:
"As outlined in the previous subsections, the TESSA architecture consists of various connected components acting asynchronously.

In order to asses its performance, it is convenient to consider it as a network whose nodes are linked in a directional way, as in Fig.2. The network is characterised by delays along the links and by node internal waiting times. Some logical steps of the flux diagram of Fig.1 can be remapped to the network of Fig.2 via the correspondence provided in Tab.1. The nodes of Fig.2 were chosen because they correspond to the places where it is presently possible to perform time measurements. The names of the nodes correspond to the name of the TESSA tier (cp. Fig.1 of NHESSD) where measurements are taken.

The performance of the VISIR operational system can then be assessed by comparing time measurements at various nodes. The experimental activity was carried out ensuring that the computational cluster was nearly idle. Computations of routes up to 500 NM were requested from the UI. For each given route length, the same departure and arrival were employed in up to 10 routing jobs during test sessions occuring on three different calendar dates. The experimental protocol also included recording

time-stamps at the various nodes of the operational infrastructure of Fig.2.

In Fig.3 we display data collected for both the motorboat (panels a,b) and sailboat (panels c,d) VISIR jobs. In the left panels we display the "waiting times" $\eta(Q) = t(Q_2) - t(Q_1)$, with $Q = \{U, S, C\}$ (see Fig.2), and $\tau(m)$, the duration of the matlab job. The measurement of the $\eta(Q)$ times, being differences of absolute times at a given tier of the TESSA infrastructure (cp. Fig.2 and Tab.1), does not require clock synchronization among the asynchronous system tiers.

For longer routes, $\eta(Q)$ is dominated by $\tau(m)$, that, with respect to route length $L$, scales as a power law with an exponent of about 3 (cp. Tab.2). The relation between $L$ and the number $N$ of grid points in the selected bounding box (see Sect.3.5 of NHESSD) depends on the aspect-ratio of the latter. For a squared box, $L \sim \sqrt{N}$, for a more elongated one, $L \sim N$. Consequently, the fitted exponent also implies a power-law dependence on $N$ with an exponent comprised between 1.5 and 3. This is compliant with the fact that a quadratic trend with respect to $N$ was fitted in (Mannarini et al., 2016). This in turn agrees with the theoretical performance of Dijkstra's algorithm (Sect.2.2 of NHESSD) in its implementation without any specific data structures. Thus, the performance of the VISIR operational system for long routes mirrors that of the model for ship route optimization.

For shorter routes however, it is apparent that the values of $\eta(Q)$ at the various system layers differ by several seconds. In order to better explore the performance of the VISIR operational system, we subtract the $\tau(m)$ dominant contribution, defining the excess times $\delta(Q) = \eta(Q) - \tau(m)$. Furthermore, we display $\delta(Q)$ vs. a proxy of the size of the payload transferred from the CDAM up to the level of the UI. This corresponds to the sum of the two files containing the geodetic and the optimal route. This way, the pictures displayed in the right panels of Fig.3 are obtained for the motorboat and sailboat cases respectively.

Starting from the CDAM $\delta$ data (blue markers and lines), a nearly constant trend is

observed for both motor- and sailboat tests. We find $\delta(C) \approx 2$ seconds. Between the $C_2$ and $C_1$ nodes two distinct processes occur on the CDAM: the matlab job is submitted via slurm and, upon completion, the results are uploaded using the callback URL mechanism (cp. Sect.3.2 of NHESSD) for making them available to the SSA platform. In order to assess the contribution from the startup submission time of a slurm/matlab job, we separately tested that the duration of a slurm task that just submits a void matlab job is between 1.4 and 2.0 seconds. Most of the $\delta(C)$ time should be ascribed to this irreducible slurm duration. $\delta(C)$ is independent of payload size, as a consequence of the high-speed internet connection of the computing facility hosting the CDAM.

For the SSA $\delta$ data (red markers and lines), again a nearly constant trend is observed for both motor- and sailboat job tests. We find $\delta(S) \approx 3$ seconds. Despite the name (arising from the TESSA tier where the measurements are carried out), $\delta(S)$ contains, besides the SSA-CDAM bidirectional delays, mainly CDAM tasks, and namely: the preliminary validation of the job request parameters (occuring at the CDAM Gateway), the setup of the remote environment on the supercomputing facility (CDAM Gateway), the namelist production (CDAM Launcher), the processes leading to $\delta(C)$ (CDAM Launcher), the creation of an output json file containing both the geodetic and the optimal route. With the exception of the first and the last, these processes are independent of payload size and, thanks to the fair internet bandwidth available at the SSA platform location, they result in a nearly constant $\delta(S) - \delta(C) \approx 1$ sec.

Moving to the UI $\delta$ data (black markers and lines), we note first of all that, for a given route length, the sailboat payload is smaller than the motorboat one. This is due to the fact that a smaller number of fields is stored in the sailboat payload (e.g. the flags of the safety constraints are just available for the motorboat case, see Sect.2.3 of NHESSD). Furthermore, the UI measurements are characterized by a large variability (error bar size) among the different tests. Both motorboat and sailboat data show a linear trend with respect to route length. The intercept of the linear fits ($e$ parameter in Tab.2) is

located at about 7 seconds. In order to understand these results, it should be recalled that $\delta(U) - \delta(S)$ includes, besides the UI-SSA bidirectional delays, the waiting time $\tau(p)$ due to the polling interval $t_{\text{poll}}$ of the client application (Sect.3.4 of NHESSD). The polling mechanism can be considered as a normally closed gate, opening instaneously any $t_{\text{poll}}$ time units. Thus, a signal arriving at the gate at a time $\sigma$ cannot pass before the additional delay $\tau(p)$, given by:

$$\tau(p) = t_{\text{poll}} - \text{mod}(\sigma, t_{\text{poll}}), \tag{1}$$

is elapsed. Eq.1 defines a piece-wise linear function of $\sigma$. In the VISIR case $\sigma$ includes both deterministic and stochastic (due to internet bandwidth and computing resources availability) processes, as described above. Thus, it is convenient to estimate the average of $\tau(p)$, given by

$$\overline{\tau(p)} = \langle \tau(p) \rangle_\sigma = \frac{1}{2} t_{\text{poll}} \tag{2}$$

In VISIR case, $t_{\text{poll}} = 5$ sec. Thus, we find that, on the average, it must be $\delta(U) - \delta(S) \geq t_{\text{poll}}/2 = 2.5$ sec. For this reason, we display in Fig.3b,d the line $\delta(S) + t_{\text{poll}}/2$ (dashed grey line) and, as expected, we find that $\delta(U)$ is never smaller than this reference.

Finally, we discuss the slope found for the linear fits of $\delta(U)$ (Fig.3b,d). We note that it is different for the motorboat and sailboat cases (Tab.2). We recall that the $\delta$ signals displayed are not affected by the VISIR model internal time $\tau(m)$, that in fact differ between motorboat and sailboat computations, as shown by the values in Tab.2. Thus, the reason for the different slopes must be searched either in the internet connection or in some UI processing. The error bars in Fig.3b,d and the uncertainty in the fitted parameters (Tab.2) confirm a large variability of the internet speed during the tests. However, there are also specific data checks and rendering peculiarities in the linechart (Fig.6b,c of NHESSD) of the VISIR sailboat interface. Thus, at the moment we cannot rule out any of the possible explanations. Both are likely to be at play, though other tests and measurements would be needed for assessing their relative weight.

**Ultimate performance limit of VISIR**

The experimental findings above and their analysis enable us to assess the ultimate performance limit of the VISIR technological infrastructure.

There are limiting factors of two types: on the one hand, there are parameters (such as $t_{poll}$ and the slurm duration for sumbitting a job) that could be tuned for obtaining some minor improvement (less than 5 seconds in the best case). On the other hand, there are larger penalties paid to the present architecture representing the actual bottlenecks of performance, namely:

a) the computational cost for generating the ship routes;

b) the internet-based comunication among asyncronous system components.

In particular, a) dominates the UI waiting time for long routes (cp. panels a,c in Fig.3) while b) is the limiting factor for the duration of the short routes.

Fixing item (a) corresponds to a situation where there are dedicated computational resources and negligible competition within the supercomputing facility and the performance of the computing algorithm has significantly improved. It can be simulated by subtracting the duration $\tau(m)$ of the matlab job, as we did in Fig.3b,d.

If further item (b) were fixed by a fast link (optical fiber or so), the ultimate limit for the VISIR user waiting time would be given by $e + t_{poll}/2$, where $e$ is the intercept of $\delta(S)$ (cp. Tab.2). Using current values for both SSA $e$ and $t_{poll}$, this would imply a total waiting time of about 6 seconds after a route request is submitted from the user interface."

B8) On P12, row 370 insert:
"We analyzed in detail the performance of the infrastructure enabling VISIR operations (Sect 4.2). In general, the CDAM and the SSA platform appear to communicate quite

efficiently, while the client internet connectivity may act as a bottleneck. Though this can easily be improved for terrestrial accesses, it could represent an even more severe issue in case of offshore use of VISIR. This is why a caching system for the imagery of the environmental fields could be developed for using the system in case of poor internet connectivity. The other bottleneck outlined in Sect.4.2 is the performance of the ship routing model and it is already being addressed by an ongoing research activity at CMCC. The ultimate performance limit of the present architecture is a user waiting time of about 6 seconds."

B9) On P12, row 379 remove:
"Also, a caching system for the imagery of the environmental fields could be a viable solution for using the system in case of poor internet connectivity."

B10) replace Fig.3 of NHESSD with Fig.1 of this comment.

B11) replace Fig.4 of NHESSD with Fig.2 ("Graph Structure..") of this comment.

B12) replace Fig.5 of NHESSD with Fig.3 ("Performance of VISIR operational system..") of this comment.

B13) replace Tab.2 of NHESSD with Tab.1 of this comment.

B14) replace Tab.3 of NHESSD with Tab.2 of this comment.

C - *Please also revise the citation style of Mannarini et al. (2015) in the body of the*

*paper because I was not able to distinguish which one (of two in the reference list) the authors actually referred to.*

*–Authors' response:*

Surely we'll do this. In the meanwhile, the GMD-Discussions manuscript evolved into the peer-reviewed paper (Mannarini et al., 2016) and the accepted paper on sailboats has been published (Mannarini et al., 2015).

*–Authors' changes to manuscript:*

C1, C2.

C1) Using reference to (Mannarini et al., 2016)

C2) Using reference to (Mannarini et al., 2015)

[Figure]

**Table 1.** Correspondence between nodes of the VISIR system (cp. Fig.1) and steps within the Data flux diagram (Fig.3 of NHESSD). The step part (s: start, e: end) is also given when relevant ("-" otherwise).

| | $U_0$ | $U_1$ | $S_1$ | $C_1$ | $m$ | $C_2$ | $S_2$ | $p$ | $U_2$ |
|---|---|---|---|---|---|---|---|---|---|
| Data flux diagram (Fig.3 of NHESSD) step # | 1 | 5a | 4c | 6a | 6b | 6c | 7 | 5a-8 | 9 |
| step part | s | s | s | s | - | e | e | s-e | e |

**References**

Flannery, B. P., Press, W., Teukolsky, S., and Vetterling, W. T.: Numerical Recipes in FORTRAN 77: The Art of Scientific Computing, 1992.

Mannarini, G., Lecci, R., and Coppini, G.: Introducing sailboats into ship routing system VISIR, in: Information, Intelligence, Systems and Applications (IISA), 2015 6th International Conference on, pp. 1–6, IEEEXplore, doi:10.1109/IISA.2015.7387962, 2015.

Mannarini, G., Pinardi, N., Coppini, G., Oddo, P., and Iafrati, A.: VISIR-I: small vessels – least-time nautical routes using wave forecasts, Geoscientific Model Development, 9, 1597–1625, doi:10.5194/gmd-9-1597-2016, http://www.geosci-model-dev.net/9/1597/2016/, 2016.

**Table 2.** Parameters and scores for the least-square fits shown in Fig.3. For the $\eta$ and $\tau$ signals (a,c panels of Fig.3) all fits are of type $ax^b + c$, while for the $\delta$ signals (b,d panels of Fig.3) the fits are of type $dx + e$. The $b$ exponent is given with the 95% uncertainty bounds, while the $e$ intercept is given with the estimated variance (among brackets). Least-$\chi^2$ linear fits (Fig.3b,d) employ formulas provided in (Flannery et al., 1992, Chap.15) for the coefficients of the linear regression and their uncertainties.

| | | | [units] | motorboat | sailboat |
|---|---|---|---|---|---|
| UI | $\eta$ | $a$ | [s·NM$^{-b}$] | $3.9 \cdot 10^{-7}$ | $2.4 \cdot 10^{-5}$ |
| | | $b$ | [-] | 3.3 (2.6 - 3.9) | 3.0 (2.5 - 3.5) |
| | | $c$ | [s] | 19.8 | 14.1 |
| | | $R^2$ | [%] | 98.5 | 99.5 |
| | $\delta$ | $d$ | [kB/s] | 90(100) | 30(18) |
| | | $e$ | [s] | 7.1(1.2) | 6.3(0.7) |
| | | $R^2$ | [%] | 80.6 | 59.3 |
| SSA | $\eta$ | $a$ | [s·NM$^{-b}$] | $9.3 \cdot 10^{-7}$ | $1.8 \cdot 10^{-6}$ |
| | | $b$ | [-] | 3.1 (2.6 - 3.7) | 3.0 (2.5 - 3.5) |
| | | $c$ | [s] | 14.5 | 10.5 |
| | | $R^2$ | [%] | 98.3 | 99.6 |
| | $\delta$ | $e$ | [s] | 3.2(0.2) | 3.1(0.2) |
| | | $R^2$ | [%] | 74.1 | 94.2 |
| CDAM | $\eta$ | $a$ | [s·NM$^{-b}$] | $1.1 \cdot 10^{-6}$ | $2.0 \cdot 10^{-6}$ |
| | | $b$ | [-] | 3.1 (2.6 - 3.7) | 3.0 (2.5 - 3.5) |
| | | $c$ | [s] | 12.6 | 8.6 |
| | | $R^2$ | [%] | 99.6 | 99.6 |
| | $\delta$ | $e$ | [s] | 2.1(0.1) | 2.1(0.1) |
| | | $R^2$ | [%] | 98.7 | 94.8 |
| matlab | $\tau$ | $a$ | [s·NM$^{-b}$] | $1.7 \cdot 10^{-6}$ | $3.7 \cdot 10^{-6}$ |
| | | $b$ | [-] | 3.0 (2.5 - 3.6) | 2.9 (2.4 - 3.4) |
| | | $c$ | [s] | 9.3 | 5.5 |
| | | $R^2$ | [%] | 97.9 | 99.5 |

**Fig. 1.** Data flux diagram for the operational system of VISIR.

[Figure]

**Fig. 2.** Graph structure of the TESSA system for running DSSs like VISIR. The nodes refer to points of time measurements.

[Figure]

**Fig. 3.** Performance of VISIR operational system, distinguishing between motorboat (a,b) and sailboat mode (c,d).

---

## Author Comment (AC2) · 23 Jun 2016

(In the following "NHESSD" stands for the paper under Review. When not specified, all other references to equations, figures, and tables are relative to the present document.)

**General comments**

*The authors present the software and operational architecture of a DSS for safe and efficient navigation. There is no doubt that the work is of high standard and involves large-scale operational modeling of several physical parameters, combined with a multi*

[Figure]

*layer interface for post-processing and visualization. I also found that the online plat-
form is of good quality and rather intuitive. Finally the paper is well written [..] .*

*–Authors' response:*
We thank the Referee for the appreciation of the model, the operational system, the
user interface, and the way the manuscript is written.

*[..] but I am afraid I cannot recommend publication without major changes. My main
concern is about the contents of the paper, and the selection of the journal, even though
it is submitted as part of a special issue on situational sea awareness.*

*–Authors' response:*
First of all, we find it useful to recall official information about the NHESS journal and
this special issue.

The scope of NHESS includes [1]:

"[..] the design, implementation, and critical evaluation of mitigation and adaptation strategies
to reduce the *impact of hazardous natural events on human-made structures* and infrastructure,
to reduce vulnerability and to increase resilience of individuals and societies; [..] "

The topic of the special issue in which we inserted the manuscript contains the follow-
ing[2]

"[..] The key is to *develop information technologies* capable of disseminating user-oriented in-
* * *
[1]http://www.natural-hazards-and-earth-system-sciences.net/about/aims_and_scope.html
[2]Before removal, it was published at http://www.natural-hazards-and-earth-system-sciences.net/special_issues/
schedule.html#4. We reproduce a screenshot of the original text in Fig. 1.

formation that is based upon rapidly changing *data from wind, waves* and current forecasts. Situational sea awareness is useful for all maritime operations, from *ship routing* to search and rescue, offshore hydrocarbon exploration and fisheries, where the knowledge of the environmental conditions is required to carry out *operations at sea safely and efficiently*. The environmental data of interest are the meteo-oceanographic analyses and forecasts as well as the *related positions of objects* and substances in the whole water column."

We think that the highlighted text proves that our manuscript is highly relevant to the scopes of both this journal and this special issue. We will elaborate more in the specific comments below.

**Specific comments**

**A** - *The paper is focusing on the software platforms and technology used to calculate and visualize optimal routes. Therefore it feels more like a technical report or manual written by a software developer and not like a manuscript prepared for a natural hazards journal. No natural processes are mentioned, while there is no theory, no initial hypothesis, no analysis and no results.*

*–Authors' response:*
Targeted services based on operational outputs from geophysical models represent a new paradigm of knowledge. For operational use by decision makers with a limited knowledge of the underlying natural processes, these services require technological tools of increasing complexity and robustness (Hey et al., 2009). Furthermore, the reproducibility and the degree of objectiveness of these services critically depend on the traceability of the computational process and the quality of the computer codes behind

it (Weintrit et al., 2013), (Mannarini et al., 2016a). This is why our NHESSD, with its declared emphasis on the technological aspects of the infrastructure for providing an operational service for the safety and efficiency of navigation, fits into the scope of the NHESS journal. In fact, such a scope includes not just natural processes but also their impact on human-made structures[3]. Motor vessels and sailboats are examples of such structures and the natural processes impacting on them are the meteo-marine conditions, represented in our case by the sea-state and wind forecasts. Our NHESSD also contains an essential description of the VISIR ship routing model (P3,P4 of NHESSD) which is functional to a better understanding of the purpose of the technological architecture. The fact that the scientific and computational issues behind the UI are made transparent to the UI end-user is one of the main achievements of the architecture. While we believe they do not need to be documented within this paper, they are described in detail in our other papers, see e.g. (Mannarini et al., 2016b), (Mannarini et al., 2016a).

Finally, to limit the more descriptive contents of the user interface, we are going to remove Table 2 of NHESSD. Also, we point out that a true manual of the VISIR web application can be found at http://www.visir-nav.com/en/help.

–*Authors' changes to manuscript:*
A1, A2, A3, A4.
A1) On P1, row10, to insert:
"Targeted services based on operational outputs from geophysical models represent a new paradigm of knowledge. For operational use by decision makers with a limited knowledge of the underlying natural processes, these services require technological tools of increasing complexity and robustness (Hey et al., 2009). Furthermore, the reproducibility and the degree of objectiveness of these services critically depend on the traceability of the computational process and the quality of the computer codes
* * *
[3]http://www.natural-hazards-and-earth-system-sciences.net/about/aims_and_scope.html

behind it (Weintrit et al., 2013), (Mannarini et al., 2016a)."
A2) On P10, row 305, to remove reference to Tab.2
A3) On P12, row 376, to insert:
"It is important to stress that the technological infrastructure developed for running VISIR in operational mode allows an end-user to drive the execution of a state-of-the-art research model, customizing his/her route needs, vessel features, and level of safety. The fact that the scientific and computational issues behind the UI are made transparent to the UI end-user is one of the main achievements of the architecture."
A4) On P13, to remove Tab.2

**B** - *The manuscript mentions some technical aspects of a web application and even though I lack the expertise to follow and evaluate the system architecture, the description looks rather vague so it doesn't really provide the reader with a clear idea of how the system was built. The examples provided in the end are the most relevant part of the paper to the scope of the journal and the special issue, but no background on the route evaluation method is provided and my final impression was that the work was submitted to the wrong journal. Starting from the title, I see a weak link between 'safe and efficient navigation' and natural hazards. Acknowledging that the title is relevant to the special issue, I find that the authors should move the software aspects to the supplementary material and focus more on the natural processes involved, as well as on the best route calculation methodology.*

*–Authors' response:*
Concerning providing more background on the route evaluation method, we devoted the entire Sect.2 of NHESSD (nearly 2 pages) to summarizing main aspects of a the ship routing model. This part is better documented in other papers ((Mannarini et al.,

2016b) for motorboats, (Mannarini et al., 2015) for sailboats) also cited in NHESSD. However, we are going to stress the link between one of the examples provided and the safety checks performed by the VISIR model.

Concerning moving the software aspects to the supplementary material, in compliance with the technological scope of this special issue (Fig. 1), we still believe that it is crucial for this paper's identity to stay focused on the architectural choices made for implementing the ship routing service.

*–Authors' changes to manuscript:*
B1.
B1) On P11, row 333 to insert:
"For instance, Fig.5c shows that, for the selected route, pure loss of stability is experienced along the geodetic route, starting from positions at 110 NM from the departure place. The optimal route instead per construction does not suffer from this hazard. However, it is up to the user to individually uncheck the default flags of the safety constraints: parametric rolling, pure loss of stability, surf-riding and broaching-to."

**C** - *The title of the special issue is also the only reason I didn't recommend to reject the manuscript, but it is clear that as a coastal oceanographer I don't have the right background to review the work, therefore I would prefer not to review the revision. Finally, the best route functionality is not available in the free version of the online system, therefore I would recommend that reviewers should be provided with free online access to the full system..*

*–Authors' response:*
We thank the Referee for suggesting that we provide full access to the VISIR interface

for a more complete evaluation. After this suggestion, on 29-04-2016 we provided him/her anonymous access through the Topical Editor. The same level of access was simultaneously granted also to Referee #1. However, we are sorry that, due to the current business model, we cannot grant such level of access to all readers of the manuscript.

*–Authors' changes to manuscript:*
The Referee kindly accepted to test the operational system through the VISIR user interface, and we reacted to his/her feedback in Author's Comment #3.

**References**

Hey, A. J., Tansley, S., Tolle, K. M., et al.: The fourth paradigm: data-intensive scientific discovery, vol. 1, Microsoft research Redmond, WA, 2009.

Mannarini, G., Lecci, R., and Coppini, G.: Introducing sailboats into ship routing system VISIR, in: Information, Intelligence, Systems and Applications (IISA), 2015 6th International Conference on, pp. 1–6, IEEEXplore, doi:10.1109/IISA.2015.7387962, 2015.

Mannarini, G., Pinardi, N., and Coppini, G.: VISIR: A Free and Open-Source Model for Ship Route optimization, in: COMPIT 2016, edited by Bertram, V., pp. 161–171, Technische Universität Hamburg-Harburg, Hamburg, 2016a.

Mannarini, G., Pinardi, N., Coppini, G., Oddo, P., and Iafrati, A.: VISIR-I: small vessels – least-time nautical routes using wave forecasts, Geoscientific Model Development, 9, 1597–1625, doi:10.5194/gmd-9-1597-2016, http://www.geosci-model-dev.net/9/1597/2016/, 2016b.

Weintrit, A., Lee, S., and Alexander, L.: Software Quality Assurance Issues Related to e-Navigation, in: Marine Navigation and Safety of Sea Transportation: Advances in Marine Navigation, pp. 73–77, CRC Press, 2013.

[Figure]

**Situational sea awareness technologies for maritime safety and marine environment protection**

- Guest editors: N. Pinardi, I. Federico, A. Olita, P. Marra, and R. Archetti

- Timeline: 01 Oct 2015–15 May 2016

- More information

Situational sea awareness (SSA) is a field of applied oceanography that relies on operational oceanographic products (analyses, re-analyses and forecasts) to inform users about environmental conditions at sea, including early warning for marine extremes and hazards. The key is to develop information technologies capable of disseminating user-oriented information that is based upon rapidly changing data from wind, waves and current forecasts. Situational sea awareness is useful for all maritime operations, from ship routing to search and rescue, offshore hydrocarbon exploration and fisheries, where the knowledge of the environmental conditions is required to carry out operations at sea safely and efficiently. The environmental data of interest are the meteo-oceanographic analyses and forecasts as well as the related positions of objects and substances in the whole water column.

**Fig. 1.** Screenshot of the "Situational sea awareness technologies for maritime safety and marine environment protection" scope.

---

## Author Comment (AC3) · 23 Jun 2016

(In the following "NHESSD" stands for the paper under Review. When not specified, all other references to equations, figures, and tables are relative to the present document.)

**General comments**

*After connecting to the online platform using the user credentials provided I can provide some more feedback about the system presented. VISIR is an impressive platform with several components, while its interface is user friendly and intuitive. From that aspect I*

[Figure]

*would consider it as an example of how science could provide services to the society.*

*–Authors' response:*
We are glad that the Referee took the opportunity to evaluate more closely the VISIR application. As a matter of fact, it was both the TESSA project's and VISIR developers' aim to develop, out of the products of the fundamental research in operational oceanography, a user-friendly application for addressing societal needs.

**Specific comments**

**A** - *Some issues which could make the platform even more user-friendly are: (1) every time I was switching from one service to the other, I was asked to log in again; which in theory shouldn't be necessary;*

*–Authors' response:*
The Referee possibly refers to the fact that, starting from http://www.visir-nav.com, it is possible to switch to other services of the TESSA portfolio (Sea-Conditions, Witoil, Ocean-SAR, MarinEnvironment, Early Warning). Though they can be accessed via the same credentials used for VISIR, the user is required a to sign-on again at each service. A single sign-on for all TESSA applications is not directly related to the usability of VISIR and is thus a low priority feature in the current development roadmap of VISIR.

*–Authors' changes to manuscript:*
none.
**B** - *(2) VISIR was often not generating a route due to problems with the calculation area, automatically generated based on the origin and destination point. If I understood correctly that box sometimes can include land areas which are blocking sea access and the calculations fail. Then the user is asked to extend the bounding box something that makes the application less intuitive, and I assume there must be a way that the correct boundaries automatically so that an outcome is always generated.*

*–Authors' response:*
We are aware of the fact that the selection of the bounding box for route computation may require an extra interaction step by the end-user (see NHESSD, P9, rows 277-284). However, the bounding box is necessary, with the present algorithm and graph structure, for reducing the computational cost and thus, the waiting time for the end-user. The default bounding box may result to be inadequate for successful route computation because of one of the two reasons:

a) the marine domain encompassed by it is not connected (as correctly stated by the Referee);

b) the marine domain is connected, but there is no solution to the routing problem given the navigational safety constraints (see NHESSD, P4, rows 102-107)

Leaving to the end-user the choice of if and how to resize the bounding box certainly introduces a degree of subjectivity in the final results of the route computation. However, it is our experience that, after a few trials, the user easily learns how to do it in a conservative and still effective way. In fact, it is sufficient to obtain a result and then submit a new route with a larger bounding box; if there is no change in the results, convergence has been achieved.
Unfortunately, it was not possible during the frame of TESSA project to develop an automated way to resize the bounding box, and this is left to future improvements.

*–Authors' changes to manuscript:*
B1.
B1) On P9, row 282, to insert:
"Leaving to the end-user the choice of if and how to resize the bounding box certainly introduces a degree of subjectivity in the final results of the route computation. However, it is our experience that, after a few trials, the user easily learns how to do it in a conservative and still effective way. In fact, it is sufficient to obtain a result and then submit a new route with a larger bounding box; if there is no change in the results, convergence has been achieved."

**C** - *Apart from the application which is impressive, I insist on my previous recommendation that a manuscript published in NHESS should present, evaluate and discuss the scientific approach the routes are calculated, rather than the IT systems and environments the web interface was built.*

*–Authors' response:*
For this comment, we refer to our previous answer to the Referee: see Author's Comment #2.

*–Authors' changes to manuscript:*
none.